# The Activation of Nrf2/HO-1 by 8-Epi-7-deoxyloganic Acid Attenuates Inflammatory Symptoms through the Suppression of the MAPK/NF-κB Signaling Cascade in In Vitro and In Vivo Models

**DOI:** 10.3390/antiox11091765

**Published:** 2022-09-07

**Authors:** Shakina Yesmin Simu, Md Badrul Alam, Sun Yeou Kim

**Affiliations:** 1College of Pharmacy, Gachon University, No. 191, Hambakmoero, Incheon 21936, Korea; 2Department of Food Science and Biotechnology, Graduate School, Kyungpook National University, Daegu 41566, Korea; 3Food and Bio-Industry Research Institute, Inner Beauty/Antiaging Center, Kyungpook National University, Daegu 41566, Korea

**Keywords:** 8-epi-7-deoxyloganic acid, antioxidant, anti-inflammation, MAPKs, NF-κB, Nrf-2, HO-1

## Abstract

In this study, we examined the ameliorative effects of 8-epi-7-deoxyloganic acid (DLA), an iridoid glycoside, on oxidative stress and inflammation in both LPS-stimulated macrophages and mice with carrageenan-induced inflammation. DLA decreased oxidative stress through the up-regulation of heme oxygenase-1 (HO-1) via the activation of nuclear factor erythroid 2-related factor 2 (Nrf2), leading to the suppression of reactive oxygen species (ROS) and nitric oxide generation (NO). In addition, DLA inhibited the activation of mitogen-activated protein kinases (MAPKs) and the nuclear factor kappa-light-chain-enhancer of activated B cells (NF-κB) pathway, resulting in a decreased production of the proinflammatory cytokines tumor necrosis factor-α (TNF-α) and interleukin-1β (IL-1β) and -6 (IL-6), as well as of monocyte chemoattractant protein-1 (MCP-1). In addition, DLA effectively inhibited the generation of nitric oxide (NO) and prostaglandin E2 (PGE2) by inhibiting the expression of the upstream genes inducible nitric oxidase (iNOS) and cyclooxygenase-2 (COX-2). DLA demonstrated powerful anti-inflammatory and antioxidant properties and thus appears as an intriguing prospective therapeutic treatment.

## 1. Introduction

Oxidative stress (OS) refers to an imbalance between the creation of reactive oxygen species (ROS) and the availability of endogenous antioxidants to neutralize them [1,2]. An overabundance of ROS can damage lipids, proteins, and DNA in biological systems. ROS directly damage biomolecules or modify proteins and genes, activating inflammatory signaling cascades that lead to diseases such as atherosclerosis, neurodegenerative diseases, obesity, diabetes, dyslipidemia, and cell injury and death [3].

Inflammation is the body’s physiological response to tissue injury, infection, or chemical irritation [4]. Infection activates macrophages, fibroblasts, mast cells, and neutrophils [5]. The activation of macrophages plays a key role in the evolution of numerous inflammatory illnesses through the release of prostaglandin (PG), nitric oxide (NO), interleukin-1β (IL-1β) and -6 (IL-6), tumor necrosis factor-α (TNF-α), monocyte chemoattractant protein-1 (MCP-1), and ROS [6]. Inhibiting pro-inflammatory mediators and cytokines in activated macrophages might help in the treatment of inflammatory diseases. Inflammation signaling involves multiple mechanisms. By interacting with their promoter regions, nuclear factor kappa-light-chain-enhancer of activated B cells (NF-κB) modulates the expression of cyclooxygenase-2 (COX-2), inducible nitric oxidase (iNOS), and other pro-inflammatory cytokines in LPS-stimulated macrophages [7]. In dormant cells, the endogenous inhibitor protein κBα (IκBα) locks NF-κB into the cytoplasm. LPS stimulation phosphorylates IκBα, initiating the ubiquitin-dependent IκBα breakdown in the proteasome, causing NF-κB to translocate swiftly into the nucleus and activate genes [8]. Mounting evidence suggests that the MAPK signaling cascade, consisting of p38 mitogen-activated protein kinases (p38), c-Jun N-terminal kinases (JNK), and extracellular signal-regulated kinases (ERKs), stimulates NF-κB transcription factor and regulates mammalian inflammation [9]. Moreover, during inflammation, cells experience recurrent oxidative stress. OS and inflammation drive the pathophysiology of many illnesses [10]. Nuclear factor erythroid 2-related factor 2 (Nrf2) and Kelch-like ECH-associated protein 1 (Keap1), implicated in the antioxidant system, play a vital role in inflammatory illness and oxidative stress [11]. When activated, Nrf2 translocates into the nucleus, binds to antioxidant response element (ARE)-containing promoter regions, which results in the expression of many phase II genes such as hemeoxygenase-1 (HO-1), and inhibits the LPS-stimulated generation of reactive oxygen species (ROS) and pro-inflammatory cytokines in activated macrophages [12].

Antioxidants and synthetic non-steroidal anti-inflammatory medications (NSAIDs) are used to treat oxidative damage and acute and chronic inflammation. Long-term exposure to these synthetic small chemicals can cause gastrointestinal problems [13]. As a result, developing novel bioactive compounds is of tremendous interest. Plant secondary metabolites have been used as remedies for centuries. Iridoids and their glycosides are found in Apocynaceae, Lamiaceae, Loganiaceae, Rubiaceae, Scrophulariaceae, and Verbenaceae [14]. They have cardiovascular, hepatoprotective, hypoglycemic, antimutagenic, antispasmodic, anti-tumor, antiviral, immunomodulatory, and purgative properties [15,16,17,18]. Additionally, a number of iridoid glucosides, including aucubin, loganin, morroniside, and geniposide, have the ability to activate the Nrf2/HO-1 signaling cascade, protecting against neurotoxicity caused by hydrogen peroxide in SH-SY5Y cells, inflammation caused by LPS in RAW 264.7 cells, and osteoblast injury caused by cadmium [19,20,21,22].

The molecule 8-epi-7-deoxyloganic acid (DLA), a monoterpenoid iridoid having hydroxy and methyl groups at positions 1 and 7, is found in *Rauvolfia serpentina, Uncaria tomentosa,* and *Hydrangea macrophylla* [23,24,25]. DLA possesses anti-inflammatory properties by inhibiting iNOS and NF-κB in LPS-stimulated RAW264.7 cells [25,26]. However, the mechanism of action of DLA against inflammation is still unknown. In this study, we examined DLA anti-inflammatory properties and molecular mechanisms of action in LPS-stimulated macrophages and in mice with carrageenan-induced inflammation.

## 2. Materials and Methods

### 2.1. Chemicals and Reagents

The molecule 8-epi-7-deoxyloganic acid (purity > 97%) was provided from the natural product chemistry lab, Jahangirnagar University, Savar, Bangladesh. LPS (from *Escherichia coli* 055:B5), dimethyl sulfoxide (DMSO), indomethacin, and hydrogen peroxide (H_2_O_2_) were supplied by Sigma-Aldrich (St. Louis, MO, USA). Invitrogen-Gibco supplied antibiotics (streptomycin and penicillin) as well as Dulbecco’s modified Eagle’s medium (DMEM) and FBS (Life Technologies, Grand Island, NY, USA). Biolegend (San Diego, CA, USA) supplied the ELISA kits for mouse tumor necrosis factor-α (TNF-α), interleukin-1β (IL-1β) and -6 (IL-6),and monocyte chemoattractant protein-1 (MCP-1). All primary antibodies (Appendix A) were obtained from Abcam (Cambridge, MA, USA), Cell Signaling (Boston, MA, USA), or Santa Cruz Biotechnology (Dallas, TX, USA). Protein-Tech delivered anti-rabbit or anti-mouse IgG conjugated with horseradish peroxidase. (Boston, MA, USA). Bioneer (Daejeon, Korea) supplied all PCR primers (Appendix A).

### 2.2. Cell Viability and Cellular ROS Production Assay

In a 96-well plate, RAW 264.7 cells (ATCC, Manassas, VA, USA) (2 × 10^5^) were cultured at 37 °C in 5% CO_2_ for 24 h, then DLA (12.5–5 μM) was administrated to the cells, which were incubated for 20 h. Cell viability was then measured using 3-(4,5-dimethylthiazol-2-yl)-2,5-diphenyl-2H-tetrazolium bromide (MTT) [27]. The cellular reactive oxygen species (ROS)-scavenging ability of DLA was determined utilizing an oxidant-sensitive fluorescent probe, 2′-7′dichlorofluorescin diacetate (DCFH-DA), in accordance with a previously described approach [28].

### 2.3. Measurement of NO, PGE_2_, TNF-α, IL-1β, IL-6, and MCP-1

DLA (12.5, 25, and 50 μM) was added to RAW 264.7 cells for 24 h with or without LPS (500 ng/mL). Then, using the Griess reagent, nitric oxide (NO) concentrations in the cell culture supernatants were determined [13]. ELISA was used to measure the levels of PGE_2_, TNF-α, IL-1, IL-6, and MCP-1 following the manufacturer’s instructions.

### 2.4. Real-Time Reverse Transcription-Polymerase Chain Reaction Analysis

The QuantiTect SYBR Green reverse transcription (RT)-polymerase chain reaction (PCR) kit was used to isolate total cellular RNA from cultivated cells (Qiagen, Valencia, CA, USA). QuantiTect SYBR Green RT-PCR Master Mix and QuantiTect RT Mix were used to synthesize cDNA from 500 ng of total RNA using primers. An ABI7500 thermal cycler was used for real-time PCR (Applied Biosystems, Foster City, CA, USA). The quantity of glyceraldehyde-3-phosphate dehydrogenase (GAPDH) mRNA was used to standardize the data [29].

### 2.5. Transfection of Small Interfering RNA (siRNA)

RAW 264.7 cells were seeded (2 × 10^5^ cells/mL) into each well of a 6-well plate and allowed to grow for 24 h. The cells were transfected with 10–50 nM siRNA using Lipofectamine RNAiMax (Invitrogen, Carlsbad, CA, USA), in accordance with the procedure provided by the per manufacturer [13]. Santa Cruz Biotechnology was the source for both the si-Nrf2 RNAs and si-control RNAs (catalog number: SC-37049, Santa Cruz, CA, USA).

### 2.6. Protein Extraction and Western Blotting Analysis

The cells were collected and lysed in radioimmunoprecipitation assay (RIPA) buffer at ice-cold temperatures. Nuclear and cytosolic protein extracts were prepared using a nuclear and cytoplasmic extraction kit from Sigma-Aldrich Co. (St. Louis, MO, USA). The BCA protein assay kit (Pierce, Rockford, IL, USA) was utilized to determine the protein content [27].

### 2.7. Mice Model of Carrageenan (CA)-Induced Inflammation

Mice (8 weeks old; body weight 25–30 g) were procured from the Institute of Cancer Research (ICR) Samtako (Osan-si, Gyenonggi-do, South Korea) and maintained under controlled conditions (temperature: 23 ± 1 °C and humidity 55 ± 5%) with a 12 h light/dark cycle. For one week, they were given unlimited access to regular water and rodent food. The Institutional Animal Care and Use Committee (IACUC) of Kyungpook National University (KNU) granted approval for the animal experiment under the reference number KNU-2021-0110. Twenty-four mice were divided into four groups of six at random. The control group (G1) was given dH_2_O. G2 served as the control group for the CA study. G3 and 4, on the other hand, were administered DLA (20 mg/kg, p.o.) and indomethacin (10 mg/kg, p.o.) correspondingly. G3 received DLA (20 mg/kg/day, p.o.) for 4 days, while a positive control dose of indomethacin (10 mg/kg, p.o.) was given 1 h before to the CA insult. An injection of CA (1% in saline solution, 50 μL/mice) was performed subcutaneously into the mice right hind paw to induce acute inflammation. A plethysmometer (UGO BASILE; Comerio, VA, Italy) was used to measure the volume of the paw edema before and every hour for the following 6 h after the CA insult [13]. The animals were then euthanized, and the skin from their right hind legs was removed, dissected, and promptly preserved at −80 °C for qRT-PCR, ELISA, and Western blot analysis.

### 2.8. Statistical Analysis

Data are expressed as the mean ± standard deviation (SD; *n* = 3). One-way analysis of variance, followed by Tukey’s multiple-comparisons test, was performed using SPSS for Windows Version 10.07 (SPSS, Chicago, IL, USA), and *p* values < 0.05 were considered statistically significant.

## 3. Results

### 3.1. Cellular Reactive Oxygen Species (ROS) Generation and Nrf2/HO-1 Signaling

DLA (8-Epi-7-deoxyloganic acid) had no toxic effect at 12.5, 25, and 50 µM, but considerable toxicity at 100 µM (IC_50_ value 99.41 ± 1.15 μM in LPS-treated RAW 264.7 cells), as shown in Figure 1B. As a result, we conducted our experiments using the harmless concentrations. LPS may initially cause a rise in ROS. Figure 1C shows that cells stimulated with LPS contained higher levels of intracellular ROS than untreated control or DLA alone-treated cells, which were dose-dependently decreased by DLA treatment similar to what observed with gallic acid (GA), a known ROS scavenger.

The activity of nuclear factor erythroid 2-related factor 2 (Nrf2) is mostly controlled by its association with Kelch-like ECH-associated protein 1 (Keap1), a transcription factor involved in proteasome degradation [30]. In a dose-dependent manner, DLA administration drastically reduced Keap-1 expression by five-fold at 50 μM (Figure 1D), while Nrf2 nuclear localization was significantly enhanced by 2.8-fold and linked with lower Nrf2 levels in the cytoplasm (Figure 1E). We also studied the effect of DLA on the expression of phase II detoxifying enzymes such as heme oxygenase 1 (HO-1). The expression of HO-1 increased 4.4-fold after DLA treatment (Figure 1F). Additionally, the cells were transfected with a small interfering RNA for Nrf2 (si-Nrf2) prior to DLA treatment to show that DLA activated HO-1 via Nrf2. Si-Nrf2 significantly lowered the expression of Nrf2 protein as expected, while the addition of DLA had no additional effects (Figure 1G, relative band intensity is shown in the adjacent figure). In si-Nrf2-treated cells, HO-1 induction by DLA was likewise successfully reversed (Figure 1G, relative band intensity is shown in the adjacent figure).

Furthermore, utilizing si-Nrf2, the production of cellular ROS and nitric oxide (NO) in LPS-induced RAW 264.7 cells was measured to see if inhibiting Nrf2/HO-1 signaling lessened the inflammatory symptoms. As expected, si-Nrf2 treatment halted the inhibitory effect of DLA on cellular ROS (Figure 1H) and NO production (Figure 1I). These finding imply that DLA anti-inflammatory effects may be linked to increased HO-1 expression by altering the Keap-1/Nrf2 pathway in activated macrophages.

### 3.2. Production of Inflammatory Mediators and Proinflammatory Cytokines and Chemokines

To test the anti-inflammatory effects of DLA were investigated in LPS-treated RAW 264.7 cells by measuring the levels of NO and prostaglandin E2 (PGE2). LPS exposure significantly increased NO and PGE2 levels compared to untreated cells (column 3 of Figure 1A,B, respectively), whereas DLA treatment attenuated this trend in a concentration-dependent manner (columns 4–6 in Figure 1A,B, respectively). DLA inhibition of NO was equal to that by N6-(1-iminoethyl)-lysine, hydrochloride (L-NIL), a selective nitric oxide synthase (iNOS) inhibitor, but N-[2-(cyclohexyloxy)-4-nitrophenyl] methanesulfonamide (NS-398), a selective cyclooxygenase 2 (COX-2) inhibitor, exhibited a higher inhibitory activity than DLA with respect to PGE2 generation in LPS-induced RAW 264.7 cells.

qRT-PCR and immunoblotting were used to assess the transcriptional and translational expression of iNOS and COX-2 and determine if they regulated the generation of NO and PGE2. LPS insult significantly enhanced the mRNA (Figure 2C,D) and protein levels (Figure 2E) of iNOS and COX-2, while DLA pretreatment mitigated this tendency in a concentration-dependent manner in LPS-stimulated RAW 24.7 cells. These findings suggested that DLA reduced the expression of iNOS and COX-2 in RAW 264.7 cells, hence preventing the inflammatory response induced by LPS.

Similarly, qRT-PCR and ELISA experiments were carried out to check the effects of DLA on the production of proinflammatory cytokines and chemokines. According to Figure 3, the LPS insult dramatically increased the mRNA expression (Figure 3A–D) of tumor necrosis factor-α (TNF-α), interleukin-1β (IL-1β) and -6 (IL-6), and monocyte chemoattractant protein-1 (MCP-1), as well as their protein levels (Figure 3E–H). In LPS-treated RAW 264.7 cells, DLA treatment drastically decreased the quantity of proinflammatory cytokines and chemokines (Figure 3A–H).

### 3.3. Nuclear Factor Kappa-Light-Chain-Enhancer of Activated B Cells (NF-κB) Signaling

NF-κB, an important transcription factor, orchestrates the generation of pro-inflammatory mediators in activated macrophages [31]. Therefore, the capacity of DLA to inhibit the NF-κB signaling pathway was evaluated. LPS stimulation elevated the phosphorylation of p65 in RAW 264.7 cells, while DLA pretreatment decreased it (Figure 4A). Furthermore, as shown in Figure 4B, immunofluorescent labeling of p65 revealed that p65 was distributed throughout the cytoplasm prior to treatment. Strong nuclear p65 staining revealed that the majority of intracellular p65 was translocated from the cytoplasm to the nucleus 1 h after LPS treatment. The nuclear translocation of p65 induced by LPS treatment was, however, diminished by DLA treatment (Figure 4B). Additionally, Western blot analysis was used to determine the level of inhibitor protein κBα (IκBα) in the cytoplasm in order to investigate the effects of DLA on IκBα proteolytic breakdown. After 3 h of LPS treatment, DLA significantly reduced IκBα degradation in the cells (Figure 4C).

The phosphorylation of IκBα by IκB kinase (IKKα/β) is a crucial step in NF-κB activation. IKKα/β phosphorylates IκBα, hence promoting its ubiquitination [32]. To analyze the process leading to the inactivation of NF-κB in further depth, we examined whether DLA altered IKKα/β activation in RAW 264.7 cells. As a consequence, LPS stimulation enhanced the phosphorylation of IKKα/β, but pretreatment with DLA dramatically decreased its phosphorylation (Figure 4D). Moreover, the proteasome inhibitor MG-132 prevented IκBα degradation, demonstrating that IκBα degradation was proteasome-mediated (Figure 4E). In addition, to determine if DLA suppressed the proteasomal activity, RAW264.7 cells were treated with DLA before being cultured with a fluorogenic proteasome peptide substrate. This revealed that DLA (50 μM) significantly lowered macrophage 26S proteasome activity (Figure 4F). These findings suggest that at least a portion of the reduced degradation of IκBα by DLA was due to a diminished proteasome activity.

### 3.4. Phosphorylation of Mitogen-Activated Protein Kinases (MAPKs)

MAPK signaling pathways regulate inflammatory mediators synthesis and release by stimulating macrophages during inflammation [33]. Thus, Western blotting was used to further explain the anti-inflammatory mechanism of DLA by analyzing the phosphorylation of MAPKs (c-Jun N-terminal kinases (JNK), extracellular signal-regulated kinases (ERKs), and p38 mitogen-activated protein kinases (p38)). The phosphorylation of ERK1/2, JNK, and p38 MAPKs was achieved after LPS treatment in RAW 264.7 cells, while DLA inhibited ERK and p38 phosphorylation (Figure 5A), but not JNK phosphorylation (data not shown). In activated RAW 264.7 macrophages, DLA might impede MAPK signal transmission. To determine if MAPK signaling reduction lessened the inflammatory symptoms, we measured the quantity of NO and ROS production in LPS-insulted RAW 264.7 cells using SB239063 and U0126, selective inhibitors of p38 and ERK, respectively, and PDTC, a pharmacological inhibitor of NF-κB. Interestingly, we observed that none of the inhibitors allowed LPS to cause NO and ROS production in RAW 264.7 cells. These findings imply that the phosphorylation of MAPKs in activated macrophages, which may result in the down-regulation of NF-κB signaling cascades, may be associated with the anti-inflammatory effects of DLA.

### 3.5. Antiinflammatory Effect in a Mouse Model of Carrageenan (CA)-Induced Inflammation

The effects of DLA were then investigated in a mouse model of inflammation caused by CA. As shown in Figure 6A, DLA treatment (20 mg/kg/day, p.o.) significantly decreased the volume of paw edema caused by CA; its effects were comparable to those of indomethacin, a common NSAID.

Reactive oxygen species (ROS) as superoxide anions, hydrogen peroxide, and peroxynitrite are produced during the acute inflammatory response [34]. The biochemical parameters of the paw edema tissues, such as the levels of malondialdehyde (MDA) and the activities of superoxide dismutase (SOD), catalase (CAT), and glutathione peroxidase-1 (GPx-1), were assessed 6 h after the intrapaw injection of CA. As seen in Figure 6B, the CA insult markedly raised the MDA level, while DLA pretreatment markedly lowered the MDA level. Additionally, as anticipated, the CA insult significantly decreased SOD, CAT, and GPx-1 activities, which were thereafter restored following DLA treatment (Figure 6C).

Since cytokines have significant effects during inflammatory processes, they can be utilized as biomarkers to detect or track inflammation as it develops [35]. As shown in Figure 6D, pro-inflammatory cytokines such as TNF-α, IL-1β, and IL-6 were present at much higher levels in CA-induced mice than in sham mice. The generation of pro-inflammatory cytokines was significantly reduced by a DLA treatment administered 30 min before CA injection.

Moreover, as demonstrated in Figure 6E–G, DLA treatment (G3) drastically decreased the expression of iNOS, COX-2, IκBα, and NF-κB, while considerably boosting the levels of HO-1 and Nrf-2 proteins compared to those in CA-treated animals (G2). Our results demonstrate that DLA reduced CA-induced inflammation in mice via activating the Nrf2/HO-1 signaling pathway and suppressing the NF-kB pathway.

## 4. Discussion

Iridoids are a large and expanding class of cyclopentane pyran monoterpenes, and the majority of them are glycosides with the glucose moiety attached to the C-1 hydroxyl group [14,15]. Most glycosides are employed as lead compounds due to their modifiability and rapid absorption; their effects on several illnesses show their importance in medicinal chemistry [36]. Iridoid glycosides protect the liver, reduce inflammation, and fight tumors [16,18]. The molecule 8-epi-7-deoxyloganic acid (DLA) possesses antioxidant and anti-inflammatory effects [25], although its precise mechanisms of action remain unknown. This is the first study to investigate the anti-inflammatory properties of DLA. Our research focused on determining the effects of DLA on the regulation of nuclear factor erythroid 2-related factor 2 (Nrf2) and nuclear factor kappa-light-chain-enhancer of activated B cells (NF-κB) signaling in LPS-stimulated macrophages and in mice with carrageenan-induced inflammation.

Oxidative stress is characterized by elevated reactive oxygen species (ROS) production, which is implicated in the pathophysiology of numerous diseases which include diabetes, cardiovascular diseases, inflammation, cancer, degenerative diseases, ischemia, and anemia [37]. Inflammation involves the increased or exaggerated production of ROS by activated inflammatory and immune cells, which can cause further damage and worsen the oxidative stress (OS)–inflammation loop. Cytokines, chemokines, and growth factors released by ROS-induced signals enhance inflammation; conversely, inflammatory processes promote OS and injury by producing ROS and other oxidants, the major causative agents of atherosclerosis, cancer, and neurological disorders [38,39]. Inflammation and oxidative stress are therefore inextricably linked.

The effects of OS and its contributing components are currently a significant concern for human health. Under stressful conditions, ROS production is amplified, and endogenous enzymatic and nonenzymatic antioxidant substances are unable to handle the overload of ROS, which contributes to the pathogenesis cancer, diabetes, cardiovascular disease, neurodegenerative diseases, and numerous inflammatory conditions [40,41]. Natural substances with antioxidant qualities act as a shield against the production of free radicals and ROS and are therefore regarded as effective therapeutic agents to lessen ailments brought on by OS [42]. In this investigation, DLA effectively reduced the load of OS by reducing the production of ROS in RAW 264.7 cells caused by LPS. Prior research demonstrated that 100 μM aucubin could prevent human neuroblastoma SH-SY5Y cells from hydrogen peroxide-induced damage by scavenging ROS generation [19]. Additionally, Loganin inhibited the oxidative response by reducing the formation of ROS in RAW 264.7 cells caused by LPS [20].

Several studies show that the Nrf2 and Kelch-like ECH-associated protein 1 (Keap1)-Nrf2 signaling cascades have an important role in inflammatory illness and OS [43]. Phytochemicals and other natural compounds scavenge oxygen-free radicals and boost the activity of antioxidants [44]. Nrf2 regulates cellular antioxidant defense. Enhancing Nrf2 activity reduces acute and chronic inflammation [13,43]. Heme oxygenase-1 (HO-1), one of Nrf2 target molecules, degrades heme to produce carbon monoxide, biliverdin/bilirubin, and free iron, which reduces reactive oxygen species (ROS) generation. Inducing HO-1 improves several inflammatory disorders [28]. This study found that DLA increased the translocation of Nrf2, boosted HO-1 production, and reduced ROS and nitric oxide (NO) levels in both LPS-treated RAW 264.7 cells and mice with carrageenan (CA)-induced inflammation, which is consistent with past research. Loganic acid, an iridoid glycoside isolated from *Cornus mas*, ameliorated the antioxidant status by increasing the activity of super oxide dismutase (SOD), catalase, and glutathione peroxidase (GPx-1) and also by modulating Nrf2/HO-1 signaling [45]. Aucubin, an iridoid glucoside from *Eucommia ulmoides*, increased Keap1 proteasomal degradation, promoted Nrf2 nuclear translocation, boosted the phase II detoxifying enzyme HO-1, and alleviated oxidative stress-induced testis damage [46].

Inflammation is a host response to invasive infections or tissue injury that helps to promote tissue healing and repair by removing undesirable stimuli including infection [47]. Macrophages are the initial inflammatory regulators; inhibiting macrophage activity can reduce the inflammatory responses and prevent chronic inflammation-related illness progression [13]. Chronic inflammation increases the levels of pro-inflammatory mediators, such as inducible nitric oxidase (iNOS) and cyclooxygenase-2 (COX-2), and cytokines such as tumor necrosis factor-α (TNF-α), interleukin-1β (IL-1β) and -6 (IL-6), and monocyte chemoattractant protein-1 (MCP-1) [6,47]. iNOS and COX-2 enhance NO and prostaglandins (PGs). These are involved in multiple sclerosis, Parkinson’s and Alzheimer’s diseases, and colon cancer pathogenesis [48]. Morroniside, an iridoid glycoside from *Corni fructus*, inhibited NF-κB in db/db mice [49]. Iridoid glycosides (monotropein, deacetylasperuloside, and deacetylasperulosidic acid) from *Morinda officinalis* inhibited NO, PGE2, TNF-α, IL-1β, and iNOS and COX-2 expression in LPS-stimulated RAW264.7 cells, indicating they may be anti-inflammatory agents [36]. In LPS-stimulated RAW 264.7 macrophages and in animals with CA-induced inflammation, DLA dose-dependently suppressed iNOS, COX-2, TNF-α, IL-1β, IL-6, and MCP-1 protein and mRNA expression.

The redox-sensitive transcription factor NF-κB is a critical regulator of inflammation-induced enzymes and cytokines such as iNOS, COX-2, TNF-α, IL-1β, and IL-6, which include NF-κB binding sites in their promoters. It has gained interest as a potential novel target for the treatment of inflammatory disorders [50]. NF-κB must enter the nucleus to boost proinflammatory cytokines transcription and production [51,52]. p65, also called RelA, is the most abundant and important NF-κB transcription factor [51]. The activated IκB kinase (IKKα/β) boosts the proteasomal degradation of inhibitor protein κBα (IκBα), resulting in the release of NF-κB, which translocates to the nucleus and induces the expression of target genes [32]. DLA promoted the proteasomal degradation of IκBα via the inhibition of IKKα/β, leading to increased translocation of p65 into the nucleus, suggesting that NF-κB pathways may be involved in suppressing pro-inflammatory cytokines production in LPS-treated RAW 264.7 cells. This finding is in line with earlier studies that showed that syringopicroside from *Folium syringae* leaves [53] and Scrodentosides A–E, isolated from *Scrophularia dentata* [54] exercise their anti-inflammatory activities by inhibiting inflammatory mediators through the down-regulation of the expression of NF-κB.

NF-κB and mitogen-activated protein kinase (MAPK) may regulate pro-inflammatory cytokines production and inflammation [55]. MAPKs include c-Jun N-terminal kinases (JNK), extracellular signal-regulated kinases (ERKs), and p38 mitogen-activated protein kinases (p38) [5]. The specific signaling routes between the three MAPKs remain unknown. The current investigation showed that DLA suppressed LPS-stimulated phosphorylation of p38 and Erk1/2, which compromise hyperreactive inflammatory responses. Moreover, LPS-stimulated macrophages exposed to NF-κB and MAPK inhibitors produced less NO and PGE2, confirming that DLA limited the release of inflammatory factors via inhibiting LPS-induced NF-κB and MAPK activation.

Cumulative data suggest that the iridoid glycoside may have anti-inflammatory properties, although its aglycone component is often inert. Park et al. [56] found that utilizing β-glucosidase to break the glucosidase bond in iridoid glycosides produced H-iridoid, which suppressed the production of TNF in LPS-stimulated RAW 264.7 cells. However, the precise structure of the H-iridoid products is yet unknown. In addition, the majority of studies on iridoids continue to concentrate on elucidating their structures and biological activities, although a systematic examination of their structural types and structure–activity relationship is still lacking. To better confirm the key active functional groups and effects of iridoids and to provide adequate data support for chemical modifications and the development of innovative medications, it is required to strengthen research on the structure and function of iridoids. In addition, pharmacokinetic studies have emphasized the potential pharmacological effects of bioactive molecules/drugs. Thus, to completely appreciate the mechanisms underlying DLA anti-inflammatory action in vivo, more pharmacokinetic and pharmacodynamic studies are required.

## 5. Conclusions

The current study provides strong preliminary evidence that the iridoid glycoside 8-epi-7-deoxyloganic acid (DLA) reduced oxidative stress by blocking lipid peroxidation, enhancing primary antioxidant enzymes, and activating the Nrf2/HO-1 signaling cascade in both in vitro and in vivo models. In all models, it also exerted anti-inflammatory effects by inhibiting inflammatory mediators and their related genes, and by reducing the levels of proinflammatory cytokines and chemokines by inhibiting MAPK/NF-κB signaling. Our findings clearly support DLA as a potential novel anti-inflammatory drug; nevertheless, more systemic tests on LPS-challenged animals are required to determine its in vivo efficacy. Future research could help identify whether DLA could be used to prevent and treat inflammation-related illnesses and oxidative stress.

## Figures and Tables

**Figure 1 antioxidants-11-01765-f001:**
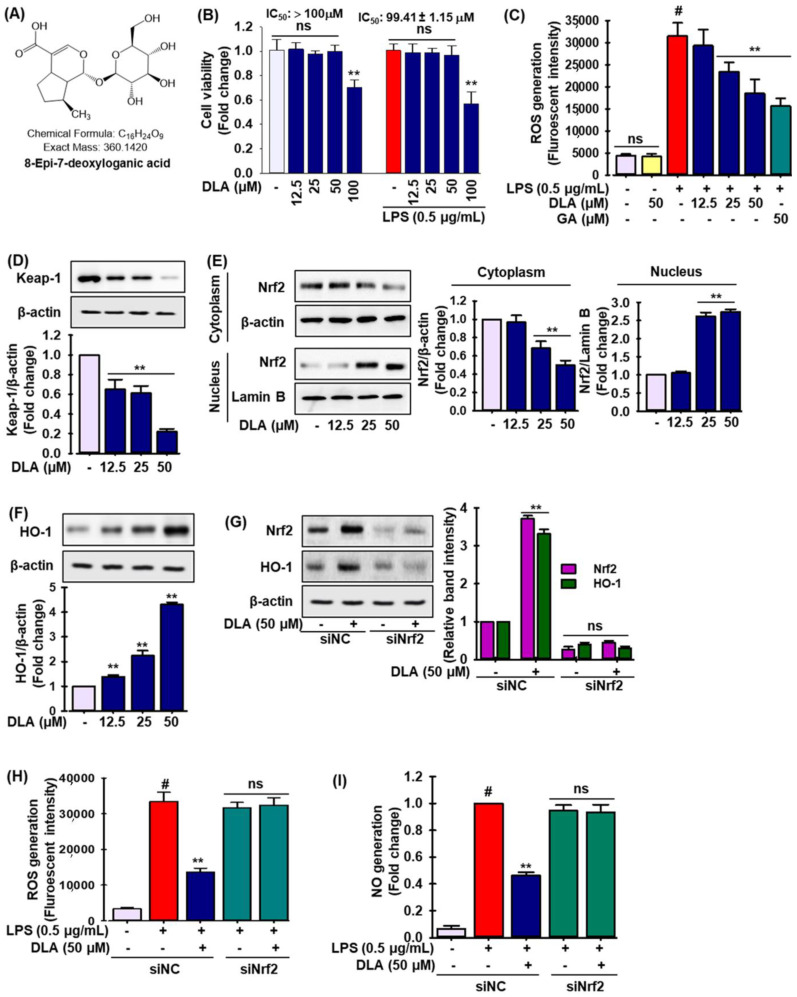
Antioxidant effects of DLA in RAW 264.7 cells. Chemical structure of 8-Epi-7-deoxyloganic acid (DLA) (**A**), A 96-well plate was seeded with cells (2 × 10^5^ cells/mL) and incubated for 24 h; Following this, the cells were exposed to DLA (12.5–50 µM) or vehicle alone for 20 h and/or for 3 h, then LPS (500 ng/mL) was added, and incubation was continued for 24 h. DLA impact on cell viability (**B**); ** *p* < 0.05 vs. treatment with vehicle and LPS alone, ns: non-significant vs. treatment with vehicle and LPS. DLA effects on intracellular reactive oxygen species (ROS) scavenging (**C**); # *p* < 0.05 vs. treatment with vehicle control; ** *p* < 0.05 vs. treatment with LPS alone, ns: non-significant vs. treatment with vehicle. Western blot analysis was used to identify the expression of Kelch-like ECH-associated protein 1 (Keap-1) (**D**), nuclear factor erythroid 2-related factor 2 (Nrf2) in both cytosol and nuclear fractions (**E**), and hemeoxygenase 1 (HO-1) (**F**); ** *p* < 0.05 vs. treatment with vehicle control. According to the procedure outlined in the materials and methods, the cells were treated with siRNA for Nrf2 (si-Nrf2) with or without DLA; The protein levels of Nrf2 and HO-1 were then determined by immunoblotting (**G**); To quantify the relative band intensities, densitometric analysis was performed; ** *p* < 0.05 vs. treatment with siNC, ns: non-significant vs. treatment with siNrf2. The production of cellular ROS (**H**) and nitric oxide (NO) (**I**) was measured; ** *p* < 0.05 vs. treatment with siNC, ns: non-significant vs. treatment with siNrf2.

**Figure 2 antioxidants-11-01765-f002:**
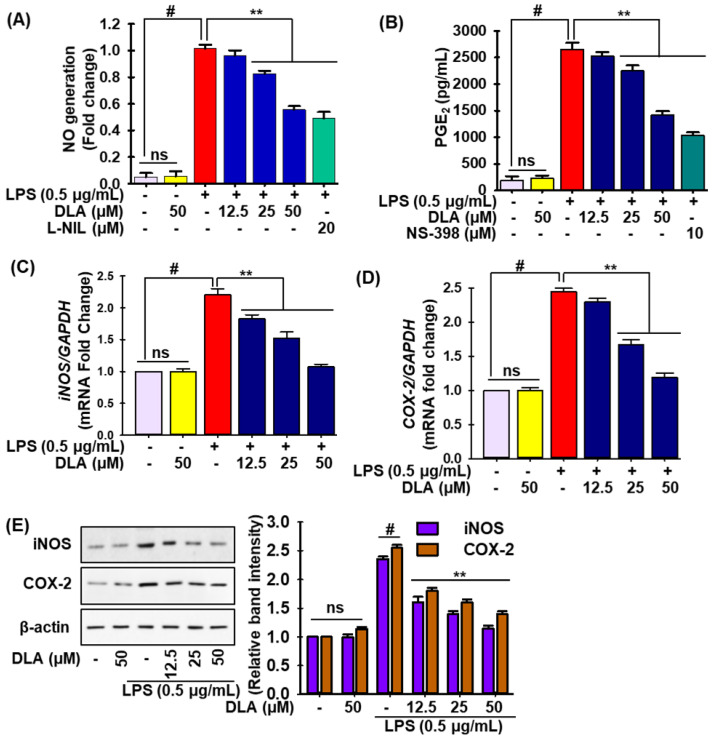
Anti-inflammatory effects of DLA. A 96-well plate was seeded with cells (2 × 10^5^ cells/mL) and incubated for 24 h. Following this, the cells were exposed to the DLA (12.5–50 µM) or vehicle alone for 20 h and/or for 3, then to LPS (500 ng/mL) for 24 h of incubation. Effect of DLA in LPS-induced RAW 264.7 macrophages on the production of NO (**A**) and prostaglandin E_2_ (PGE_2_) (**B**) and on the mRNA expression of inducible nitric oxide synthase (iNOS) (**C**) and cyclooxygenase-2 (COX-2) (**D**); protein expression of iNOS and COX-2 (**E**). To evaluate the quantification of the relative band intensities, densitometric analysis was performed; # *p* < 0.05 vs. treatment with the vehicle control; ** *p* < 0.05 vs. treatment with LPS alone. ns: non-significant vs. treatment with vehicle control.

**Figure 3 antioxidants-11-01765-f003:**
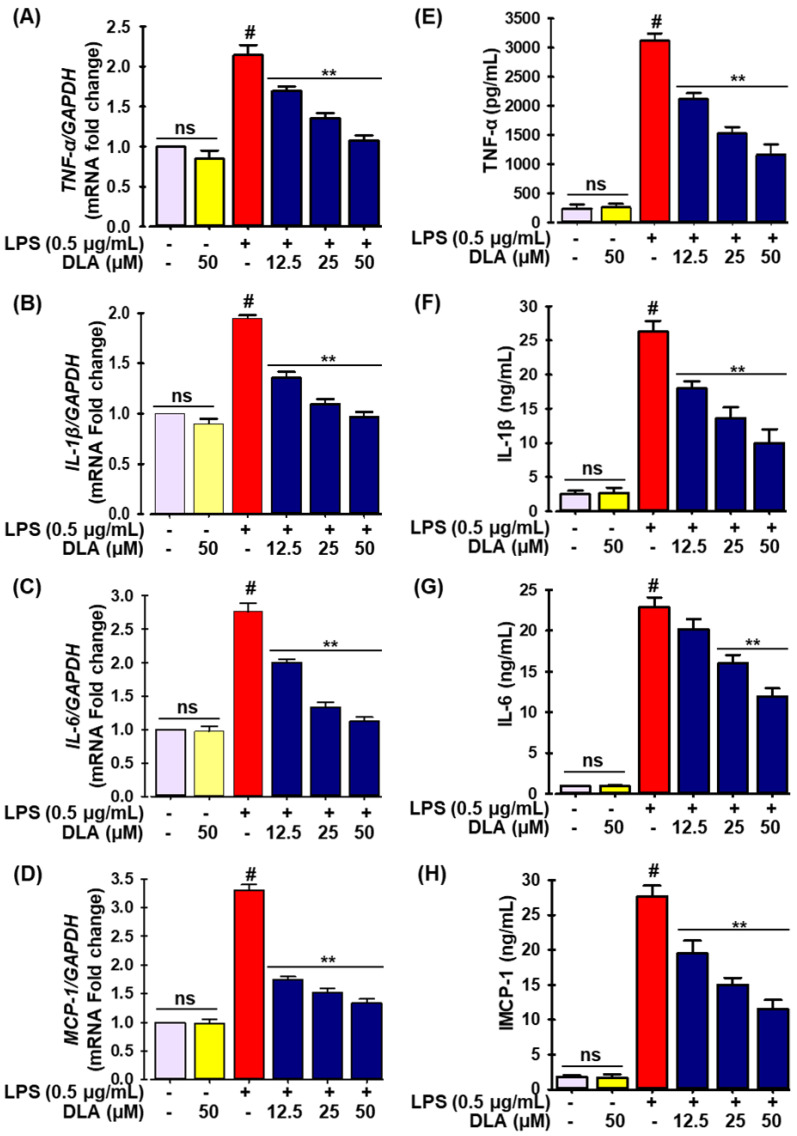
DLA impact on the production of proinflammatory cytokines and chemokine. A 96-well plate was seeded with cells (2 × 105 cells/mL) and incubated for 24 h. Following this, the cells were exposed to DLA (12.5–50 µM) or vehicle alone for 20 h and/or for 3, then to LPS (500 ng/mL), and finally they were incubated for 6 h. DLA impact on LPS-induced RAW 264.7 cells mRNA expression of tumor necrosis factor-α (TNF-α), interleukin-1β (IL-1β), interleukin-6 (IL-6), and monocyte chemoattractant protein-1 (MCP-1) (**A**–**D**) and protein expression (**E**–**H**). # *p* < 0.05 vs. treatment with vehicle control; ** *p* < 0.05 vs. treatment with LPS alone. ns: non-significant vs. treatment with vehicle control.

**Figure 4 antioxidants-11-01765-f004:**
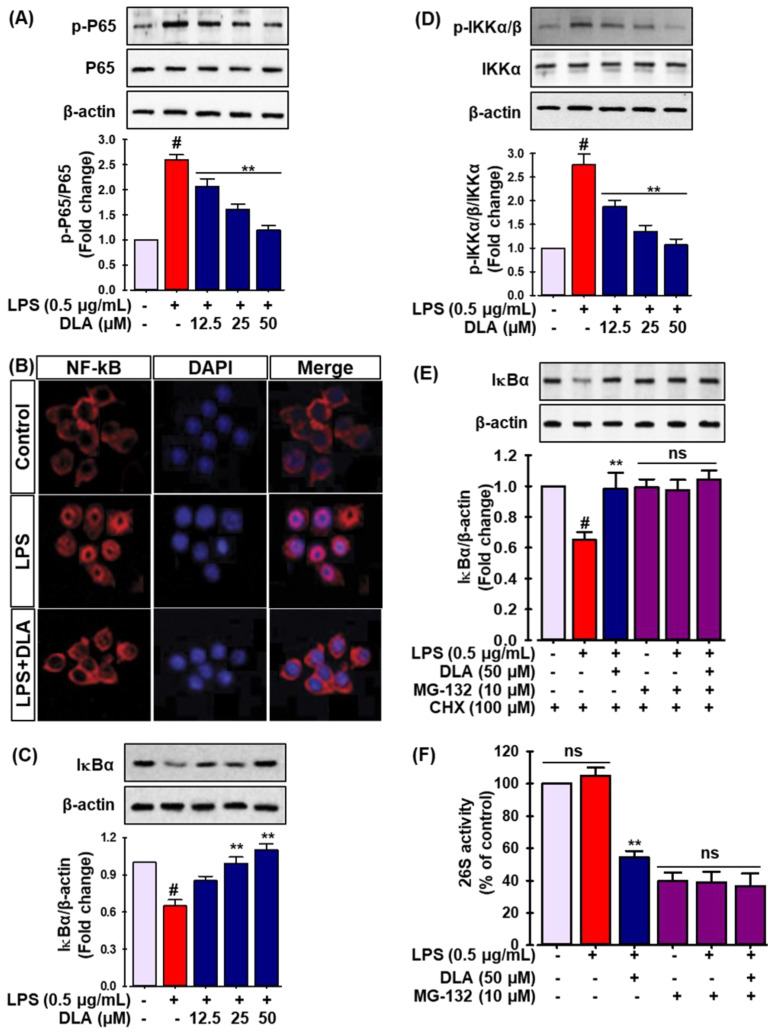
Effect of DLA on the nuclear factor kappa-light-chain-enhancer of activated B cells (NF-κB) signaling pathway. The cells were pretreated with DLA (12.5, 25, and 50 μM) for 12 h before being exposed to LPS (500 ng/mL) for 1 h. Immunoblot analysis was performed to evaluate the protein levels of p-p65 and p65 (**A**); # *p* < 0.05 vs. vehicle-treated control; ** *p* < 0.05 vs. treatment with LPS alone. The cells were fixed for immunofluorescent labeling for p65. Red represents p65; blue represents DAPI staining of the nuclei (**B**). The protein levels of inhibitor protein κBα (IκBα) (**C**), IκB kinase p-IKKα/β, and IKKα/β (**D**) were quantified by Western blot; # *p* < 0.05 vs. vehicle-treated control; ** *p* < 0.05 vs. treatment with LPS alone; Cells were treated with 500 ng/mL of LPS and 10 μM MG-132 for 30 min. Following cell harvesting, the protein level of IκB was assessed by Western blot analysis, and the relative change was determined (**E**), the cells were pre-treated with 50 μM DLA for 12 h and then incubated with 10 μM MG-132 for 1 h; After 30 min of treatment with 500 ng/mL of LPS, the proteasome activity in the cell lysates was determined (**F**); # *p* < 0.05 vs. vehicle-treated control; ** *p* < 0.05 vs. treatment with LPS alone, ns: non-significant vs. treatment with MG-132 alone.

**Figure 5 antioxidants-11-01765-f005:**
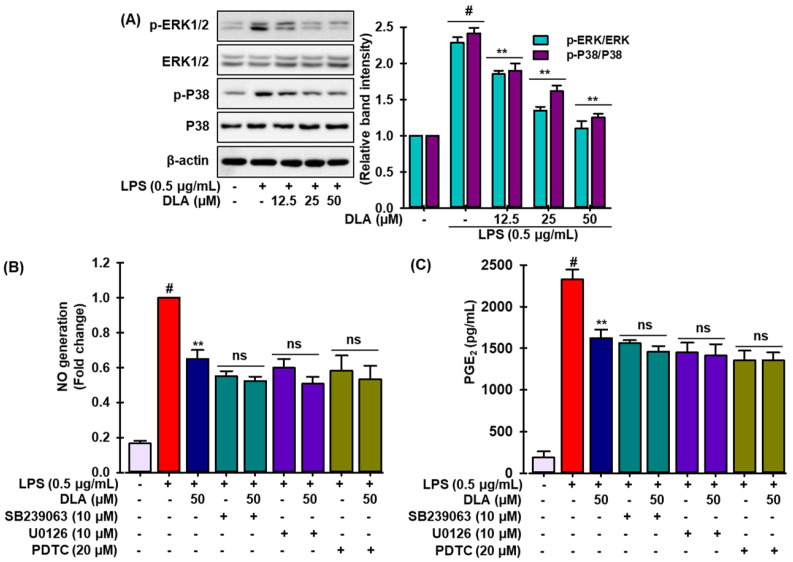
Inhibition of mitogen activated protein kinases (MAPKs) phosphorylation by DLA. A 96-well plate was seeded with cells (2 × 105 cells/mL) and incubated for 24 h. Following this, the cells were exposed to DLA (12.5–50 µM) or vehicle alone for 20 h and/or for 3 h, then to LPS (500 ng/mL), and finally they were incubated for 30 min. In LPS-stimulated RAW 264.7 cells, the effects of DLA on extracellular signal-regulated kinases (ERKs) and p38 mitogen-activated protein kinases (p38), phosphorylation were examined, and the relative change was calculated (**A**) # *p* < 0.05 vs. treatment with vehicle control; ** *p* < 0.05 vs. treatment with LPS alone. NO production (**B**) and ROS generation (**C**) in the presence of various inhibitors were determined according to the method described in the Materials and Methods section. # *p* < 0.05 vs. treatment with vehicle control; ** *p* < 0.05 vs. treatment with LPS alone, ns: non-significant vs. treatments with the respective inhibitors alone.

**Figure 6 antioxidants-11-01765-f006:**
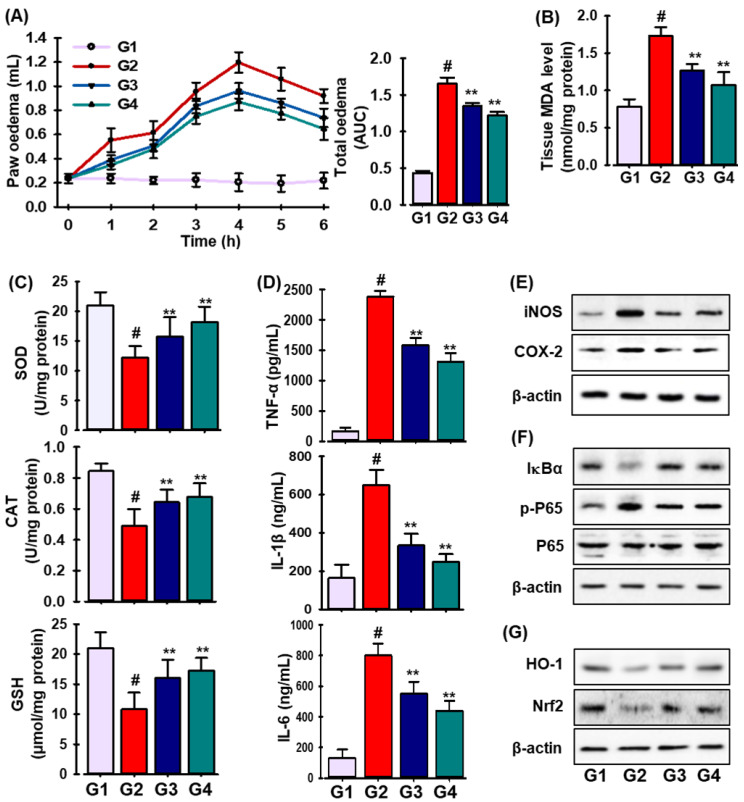
Carrageenan (CA)-induced paw edema is prevented by DLA. Every animal in every group—except G1, which served as the standard saline control—received carrageenan (50 mL/mice). As indicated in the Materials and Methods section, the paw volumes were measured 0–6 h after CA injection (**A**). Production of MDA (**B**); antioxidant enzymes (SOD, CAT, and GPx-1) activities (**C**); generation of pro-inflammatory cytokines (TNF-α, IL-1β, and IL-6) (**D**) in CA-treated paws. Protein expression levels of iNOS and COX-2 (**E**); IKBa, p-p65, and p65 (**F**); HO-1, and Nrf2 (**G**) in CA-treated mice. # *p* < 0.05 significant vs. treatment with vehicle-treated control; ** *p* < 0.05 significant vs. treatment with CA.

## Data Availability

The data presented in this study are openly available.

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
