# Peer review of "The Activation of Nrf2/HO-1 by 8-Epi-7-deoxyloganic Acid Attenuates Inflammatory Symptoms through the Suppression of the MAPK/NF-κB Signaling Cascade in In Vitro and In Vivo Models"

_antioxidants, 2022, doi:10.3390/antiox11091765_

Round 1

Reviewer 1 Report

S.Y. Simus and co-authors reported here a study of the mechanism of action of 8-Epi-7-deoxyloganic acid (DLA) against inflammation in LPS-stimulated macrophages and carrageenan-induced inflammatory mice. The obtaind results have shown that DLA reduces oxidative stress by blocking lipid peroxidation, enhancing primary antioxidant enzymes, activating the Nrf2/HO-1 signaling cascade and via in-hibiting LPS-induced NF-kB and MAPK activation.

In my opinión this is a good piece of work and the manuscript could be accepted for publication once some problems have been solved. 

1. Figure 1(A) should be revised and the stereochemistry of DLA properly represented, not only in the sugar fragment but also in the aglycone. 

2. In the citotoxicity study. Figure 1(B), the IC50 value of DLA in RAW 264.7 cells must be determined.

3. As a glycosidic derivative, its metabolic stability should be determined. Is the presence of the sugar moiety necessary for activity? The observed activity is also mantained in the aglycone form or only in the glycoside form?

Author Response

Query 1: Figure 1(A) should be revised and the stereochemistry of DLA properly represented, not only in the sugar fragment but also in the aglycone. 

Response: I appreciate your thoughtful comments very much. We corrected the DLA structure as needed. Please anticipate the change in Figure 1A.

  1. In the citotoxicity study. Figure 1(B), the IC50 value of DLA in RAW 264.7 cells must be determined.

Response: Thank you very much for your informative remarks. We calculated the IC50 value of DLA in RAW264.7 cells and placed it at the top of the updated manuscript's amendment figure 1B as well as in the text (line no. 153).

  1. As a glycosidic derivative, its metabolic stability should be determined. Is the presence of the sugar moiety necessary for activity? The observed activity is also maintained in the aglycone form or only in the glycoside form?

Response: I much appreciate your informative comments. This is one of the limitations of our research. In response to this restriction, we included these details in a separate column in the amended version of our publication. We hope our each reviewer will take our limitations into account.

“Cumulative data suggests that iridoid glycoside may have anti-inflammatory properties, although its aglycone component is often inert. Park et al. (2010) found that utilizing b-glucosidase to break the glucosidase bond in iridoid glycosides produced H-iridoid, which suppressed the production of TNF- in LPS-stimulated RAW 264.7 cells. However, the precise structure of H-iridoid products is yet unknown. In addition, the majority of studies on iridoids continue to concentrate on elucidating their structures and biological activities, although a systematic examination of their structural types and structure-activity relationship is still lacking. To better confirm the key active functional groups and effects of iridoids and to provide adequate data support for chemical modification and the development of innovative medications, it is required to strengthen research on the structure and function of iridoids. In addition, pharmacokinetic studies also emphasized the potential pharmacological effects of bioactive molecules/drugs. Thus, to completely appreciate the mechanisms underlying DLA's anti-inflammatory action in vivo, more pharmacokinetic and pharmacodynamic studies are also required. Please refer to the line no. 538-552

Reviewer 2 Report

[General]

The aim of this manuscript is to clarify the antiinflammatory effects of 8-Epi-7-deoxyloganic acid (DLA) on mouse macrophage RAW264.7 cells and carrageenan-induced inflammatory model. The authors revealed that DLA activated Nrf2/HO-1 system and reduced reactice oxygen species (ROS) and nitric oxide (NO) production. Moreover, DLA inhibited activation of signal transduction pathway (NF-κB, ERK, and p38 MAPK), and proteosome activity. The antiinflammatory effects of DLA may be important and available to treat inflammatory diseases.

The results in this manuscript is clear and consistent. However, the method of displaying data must be improved. Therefore, several corrections are needed.

[Major points]

1. The author should describe the anti-inflammatory effects of DLA in more detail in the Introduction section (line 76). What model was used and what effects were obtained in reference 25 and 26?

2. Y-axis is broken and the scale is not evenly spaced in many graphs in this manuscript. Truncated graph must not be used in bar graph. In particular, the graph with broken axis must be rejected. Such graphs lose the correct judgment about the values.

[Minor points]

1. The authors used one-way ANOVA for statistical analysis (line 147). However, ANOVA is unable to compare among groups. Although post hoc test is required to compare between two groups, the used method is not described.

2. The authors described that "p < 0.01 and p < 0.05 were considered statistically significant" (line 149). Please described the distinguish these two significance level. However, I think that only one significant level (p < 0.05) is sufficient in this manuscript.

3. The authors used horizontal bar and "ns", #, or * to show the results of statistical test in many figures. Each comparison result should be shown.

Moreover, the readers will confuse which groups the authors compared in Figure 1G, 2E, 4F, and 5. These figures should be improved.

Author Response

[Major points]

Query 1.  The author should describe the anti-inflammatory effects of DLA in more detail in the Introduction section (line 76). What model was used and what effects were obtained in reference 25 and 26?

Response: I much appreciate your thoughtful comments. We presented information regarding the anti-inflammatory characteristics of DLA. DLA had anti-inflammatory activities by reducing iNOS and NF-B activity in RAW 264.7 cells treated with LPS. Please await the revision of lines 75-76 in the updated manuscript. 

Query 2. Y-axis is broken and the scale is not evenly spaced in many graphs in this manuscript. Truncated graph must not be used in bar graph. In particular, the graph with broken axis must be rejected. Such graphs lose the correct judgment about the values.

Response: I much appreciate your informative comments. We made the necessary adjustments to all figures. Please refer to the updated figures 1 to 6. 

[Minor points]

Query 1. The authors used one-way ANOVA for statistical analysis (line 147). However, ANOVA is unable to compare among groups. Although post hoc test is required to compare between two groups, the used method is not described.

Response: Thank you very much for your informative remarks. We made the necessary changes. Please see corrected manuscript lines 146-148.

Query 2. The authors described that "p < 0.01 and p < 0.05 were considered statistically significant" (line 149). Please described the distinguish these two-significance level. However, I think that only one significant level (p < 0.05) is sufficient in this manuscript.

Response: We really appreciate your thoughtful comments. We made the necessary change.

Query 3. The authors used horizontal bar and "ns", #, or * to show the results of statistical test in many figures. Each comparison result should be shown.

Response: I appreciate your thoughtful response. The necessary adjustment has been made. All figures have been updated to reflect the new legend (Figure 1-6).

Query 4. Moreover, the readers will confuse which groups the authors compared in Figure 1G, 2E, 4F, and 5. These figures should be improved.

Response: I appreciate your thoughtful response. The necessary adjustment has been made.